# Heparan Sulfate Glycosaminoglycan Is Predicted to Stabilize Inflammatory Infiltrate Formation and RANKL/OPG Ratio in Severe Periodontitis in Humans

**DOI:** 10.3390/bioengineering9100566

**Published:** 2022-10-18

**Authors:** Roko Duplancic, Marija Roguljic, Darko Bozic, Darko Kero

**Affiliations:** 1Study Program of Dental Medicine, University of Split School of Medicine, 21000 Split, Croatia; 2Department of Oral Pathology and Periodontology, Study Program of Dental Medicine, University of Split School of Medicine, 21000 Split, Croatia; 3Department of Periodontology, School of Dental Medicine, University of Zagreb, 10000 Zagreb, Croatia; 4Laboratory for Early Human Development, University of Split School of Medicine, 21000 Split, Croatia

**Keywords:** heparan sulfate glycosaminoglycan, immunofluorescence, inflammation, periodontitis, statistical modeling

## Abstract

Since chronically inflamed periodontal tissue exhibits extracellular matrix (ECM) degradation, the possible alternative to standard periodontitis treatment is to restore ECM by supplementing its components, including heparan sulfate glycosaminoglycan (HS GAG). Supplementation of the degraded ECM with synthetic derivatives of HS GAGs has been shown to be effective for periodontal tissue regeneration in experimental animal models of periodontitis. However, the potential of HS GAG supplementation for the treatment of periodontal disease in humans is still unknown. Here, we used a statistical model to investigate the role of HS GAG on inflammatory infiltrate formation and alveolar bone resorption in humans with severe periodontitis. The model was based on data from immunofluorescence staining (IF) of human gingiva samples, and reconstruction of a subset of HS GAG -related proteins from STRING reactome database. According to predictions, increased expression of native HS GAG might stabilize the accumulation of gingival inflammatory infiltrate (represented by the general inflammatory cell marker CD45) and alveolar bone resorption (represented by Receptor Activator of Nuclear ΚΒ ligand (RANKL) and osteoprotegerin (OPG) ratio) but could not restore them to healthy tissue levels. Therefore, supplementation of native HS GAG may be of limited benefits for the treatment of sever periodontitis in humans.

## 1. Introduction

Periodontitis is one of the most common chronic inflammatory diseases. Worldwide, almost 11% of the population is affected by severe forms of the disease [1]. While many environmental and intrinsic factors play a role in the pathogenesis of periodontitis, it is well known that the inadequate host immune response to the presence of microbial biofilm on the tooth surface perpetuates the vicious cycle of inflammation and periodontal tissue breakdown. Severe forms of periodontitis are characterized by extensive degradation of periodontal tissue, which ultimately leads to the loss of the affected teeth [2,3].

Extracellular matrix (ECM) components such as heparan sulfate glycosaminoglycans (HS GAGs) and their corresponding cell surface receptors such as HS proteoglycans (HSPGs) of the syndecan (Sdc) family have long been shown to play important roles in tissue homeostasis [4,5,6]. Studies on Sdcs knockout mice (KO) have shown that the absence of individual Sdcs in tissues can promote inflammation, disrupt wound healing, and increase resistance to microbial infection [7,8]. In addition, overexpression of Sdcs has been reported in various pathological conditions [9,10]. Therefore, Sdcs may be characterized as modulators of cellular processes. Many of these effects are in turn mediated by HS GAGs, which are the major ligands of Sdcs. HS GAGs can sequester and release a variety of important regulators of cell behavior (growth factors, morphogens, inflammatory mediators). In addition, the turnover of HS GAGs is highly responsive to changes in the local microenvironment and is influenced by several enzymes that regulate the biosynthesis, modification, and degradation of HS GAGs. Total and/or conditional KOs of genes encoding these enzymes (such as EXTs, NDSTs, and HPSE1) may also influence the course of inflammation and susceptibility to microbial infection in various tissues in multiple ways [11,12,13,14].

The gold standard in the treatment of periodontitis is nonsurgical subgingival instrumentation of the affected root surfaces to remove biofilm formation, and in some cases, antimicrobials with antiseptic and antibiotic agents are used [15]. However, other approaches that might influence the host response to the bacterial load have not been adequately explored. Because chronically inflamed periodontal tissues exhibit persistent degradation of the ECM, the alternative to standard treatment is to rebuild the ECM using synthetic derivatives of ECM components. In a series of in vivo studies in rodent models of periodontitis, a synthetic HS GAG based on dextran polymers was administered to diseased animals by multiple subcutaneous injections [16]. In addition to the resolution of gingival tissue inflammation, regeneration of alveolar bone, periodontal ligament, and cementum was also reported. These results are significant for two reasons: (i) periodontal disease in the animals was induced by mono- or polymicrobial infection with periodontopathogenic bacteria, which more closely mimics the onset of periodontitis in humans compared with ligature-based models; (ii) resolution of inflammation and regeneration of periodontal tissue occurred without any attempt to control the microbial component by mechanical debridement of the affected teeth. Therefore, these results suggest that drugs based on derivatives of ECM components (such as HS GAG) may be sufficient for the complete treatment of periodontitis.

While synthetic HS GAGs have been shown to be effective in experimental animal models of periodontitis, their potential for treating periodontal disease in humans is still unknown. Using a statistical model, we aimed to predict whether changes in the bioavailability and/or structure of native HS GAG have the potential to resolve inflammation and promote alveolar bone regeneration in humans with severe forms of periodontitis. The statistical model was based on in silico co-localization of several markers from immunofluorescence staining (IF) of human gingival samples and reconstruction of the reactome of a subset of HS GAG -related proteins using the Protein Ontology database.

## 2. Materials and Methods

### 2.1. Samples Procurement

This study is based on data obtained from human gingival samples preserved as histological slides in the archival collection of the Department of Anatomy, Histology and Embryology, School of Medicine, University of Split. The samples were obtained from 40 participants, who were divided into two equally sized groups: healthy controls and periodontitis affected. The periodontitis group consisted of patients diagnosed with generalized periodontitis at stage III or IV according to the latest classification [17]. Detailed inclusion and exclusion criteria have been described previously [18]. In short, the inclusion criteria were age of at least 18 years, good general health, healthy periodontal tissue (controls) and severe periodontitis stage III and IV (periodontitis group). The exclusion criteria were the presence of systemic diseases and conditions (e.g., diabetes mellitus), long-term medication or medical history of systemic antibiotic therapy within last six months, pregnancy, alcohol and drug use, and the presence of lesions in the vicinity of the sampling area.

In short, study participants were recruited from the Department of Oral Pathology and Periodontology University of Split Hospital Centre. Clinical parameters such as probing depth, gingival recession, clinical attachment level, full-mouth plaque score and full-mouth bleeding scores were recorded. Patients were selected only after referral for clinical crown lengthening (both groups), surgical periodontal treatment, and/or tooth extraction (periodontitis group only). The samples of gingiva were stored in sealed containers with paraformaldehyde and immediately taken to the laboratory for further processing.

Each participant received, reviewed, and signed two documents: informed consent to participate in the study and consent to the collection and processing of personal data, all in accordance with the WMA Declaration of Helsinki [19], the EU General Data Protection Regulation, and Croatian laws: Health Protection Act, Health Data and Information Act, Patients’ Rights Protection Act, and Dentistry Act. 

### 2.2. Immunohistochemistry and Immunofluorescence Staining

Immunohistochemical staining (IHC) was performed on every 10th slide using hematoxylin/eosin (H/E) to verify the preservation of tissue morphology and proper alignment of tissue compartments (gingival epithelium, gingival sulcus epithelium, and subepithelial stroma). We also used the Alcian Blue staining kit (pH 1.0, mucin staining) (ab150661) (Abcam plc, Cambridge, UK), a polyvalent basic dye that binds to highly sulfated mucosubstances (including glycosaminoglycans). Staining with Alcian Blue was performed according to the manufacturer’s instructions.

The IF staining protocol from our laboratory has been described previously [20,21]. For staining with anti-HS3G10, a minor adaptation of the protocol was necessary including pretreatment of histological slides with the enzyme Heparinase III (0.02 IU/50 µL) (Seikagaku Corp, Tokyo, Japan) at 37 °C for 2 h was performed before antigen retrieval [22]. The primary and secondary antibodies used for this study are listed in Table 1.

### 2.3. Image Acquisition and Processing

Acquisition and processing of images of histological sections of gingiva have been described previously [18,22,23]. Slides were photographed using the panoramic technique at ×10 magnification with a Zeiss Axiocam 506 digital color camera (resolution 2752 × 2208 pixels (px)) mounted on a Zeiss Axio Observer inverted epifluorescence microscope (Carl Zeiss Microscopy GmbH; Jena, Germany). Equalization and automatic stitching of the individual photomicrographs (tiles) was performed in the software ZEN Blue 2.5 (Carl Zeiss Microscopy; Jena, Germany) using the “stitching tool” after all tiles had been acquired. The panoramic images were saved in the proprietary Zeiss CZI raw format and exported to the 8-bit TIFF format for further processing.

Panoramic TIFF images were processed using Adobe Photoshop 2020 (Adobe Inc., San Jose, CA, USA) as previously described [18,22]. The processed images were then used for quantification of IF signals and/or IHC staining.

### 2.4. Histomorphometry and IF Signal Quantification

The procedures for histomorphometry and quantification of IF signals from panoramic images have been described in detail previously [18,22]. Here, we focused exclusively on quantifying the spatial gradients of IF signals, i.e., determining the intensity/amount of IF signal per unit of space. The intensity/amount of IF signals is calculated as the average grey value (GC) or luminance of pixels (px) per unit of space with dimensions of 1 px (height) × number of px corresponding to the panoramic image width. The values are plotted in a top-down (T-D) 2D plots, which may contain several thousand values for each panoramic image. The T-D plots were created in ImageJ (ImageJ software, U.S. National Institutes of Health, Bethesda, MD, USA) and exported as Excel spreadsheets (Microsoft Office Excel 2016; Microsoft Corporation, Redmond, WA, USA). These were subsequently used for compatibility testing of serial sections (T-D plots of DAPI staining), in silico co-localization of IF from multiple markers, and statistical modeling. For better visual representation of the signal intensity of IF, 4-color heat maps were created. The heatmaps show the distribution of different intensity of IF signals on a scale of 0–255 px GV as follows: BLUE (10–49 px GV; low intensity), GREEN (50–149 px GV; moderate intensity), RED (150–254 px GV; strong intensity), and YELLOW (255 px GV; very strong intensity). The procedure for the making of heatmaps was described previously [24].

### 2.5. Statistical Analysis and Modeling

Only T-D plots of panoramic images of gingival sections from the periodontitis group were used for statistical modeling. To exclude background px around the histologic sections, values from the raw T-D plots were recalculated as relative values and expressed as percentages of the maximum (MAX) GV as previously described (ref.). T-D plots for each marker were then aggregated and transformed by calculating moving averages at intervals of 5000 data points. The compressed T-D plots (each containing 7000 data points) were then incorporated into a system of regression equations (first-level models) that could be used to simulate the network effects of the interactions between HS GAG, HS GAG -related factors (HSPGs and enzymes), the inflammatory infiltrate (visualized by the general inflammatory cell marker CD45 (PTPRC)), and alveolar bone degradation/regeneration (RANKL/OPG axis). The advantage of such a system is that each variable can act simultaneously as a predictor (independent variable) and outcome (dependent variable). The template for pairing the variables in the regression equation system was based on the representation of the functional protein network using the STRING reactome database [25]. Because the factors studied belong to the same functional group and are closely related, the construction of the regression equation system was divided into four modules. The main outcome variable from each module was set as the main predictor (link) for the subsequent module until all modules were closed in a loop.

Predictions from the original regression equations were calculated for each outcome variable. Predictions were made by varying the mean spatial gradient value of the predictor in 60 steps from 0 (virtual knockout—VKO) to 4-fold overexpression relative to baseline in diseased gingiva. Predictors were recorded both as the mean spatial gradient of the outcome variable and as the deviation from the mean spatial gradient of the baseline value in diseased gingiva (in percent; % change). For this purpose, another set of matrices was created. These matrices were used to calculate additional regression functions (second-level models) that were incorporated into the final regression equation system (third-level models) for predicting the expression profiles of the factors studied.

Spline (piecewise) regression (SplR) and multiple linear regression (MLR) were used interchangeably for function fitting in first-level models. The knots for SplR were set manually for each pair of variables depending on the shape of the spatial gradient curve. Segments between the knots were fitted either linearly (simple linear regression) or nonlinearly (2nd and 3rd order polynomial nonlinear regression). The decision for the type of adjustment was made according to the following guidelines: (i) known effect of the predictor on the outcome variable based on experimental data from the relevant literature dictating the type of correlation between the two (positive correlation, negative correlation, linear, nonlinear); (ii) since data in T-D plots are expressed as percentages (fractions of MAX px GV), the value of the *y*-axis intercept should be between 0 and 100; (iii) the coefficient of determination (R^2^) per segment should not be less than 0.9 to minimize the cumulative error for the model representation of each outcome variable; (iv) for very short segments up to 4 data points, linear fitting is preferred to nonlinear fitting if the *y*-axis intercept condition is met; (v) the functions included in SplR must reflect the effect of peak or floor plateaus—because the model variables represent ECM components and factors associated with ECM components that exhibit such characteristics, some degree of nonlinearity had to be introduced by SplR for each pair of variables. The second and third level models were fitted with non-linear functions (polynomial regression) only.

According to ASA Statement on Statistical Significance and *p*-values, *p*-values for regression tests were presented as calculated and should be interpreted accordingly [26]. Statistical analysis was performed in Microsoft Office Excel 2016 (Microsoft Corporation, Redmond, WA, USA). Regression matrices and calculator for prediction of expression profiles are included in the supplemental material (Appendix A).

## 3. Results

### 3.1. Expression of HS GAG and Associated Factors in Healthy Gingiva and in Gingiva of Patients Affected by Periodontitis

Healthy gingival tissue shows an abundance of highly sulfated mucosubstances throughout the subepithelial stroma compared with the subepithelial stroma of gingiva from patients with periodontitis. HS GAGs labeled by HS10E4 and HS3G10 staining are highly expressed in the epithelial compartment of both healthy and diseased gingiva, whereas their expression is sparsely distributed in the subepithelial stroma. The majority of HS GAG in the subepithelial stroma is bound to connective tissue strands and blood vessel walls (Figure 1). Different populations of stromal cells express HS GAG, including fibrocytes, vascular endothelial cells, and inflammatory cells. Similar expression patterns can be observed for the HS GAG cell surface receptors and enzymes for the synthesis and modification of HS GAG at the interface between infiltrated and noninfiltrated subepithelial stroma and alveolar bone (Figure 2). Comparison of the mean spatial gradients of HS GAG in healthy and diseased gingiva indicates different expression profiles of healthy and diseased gingiva (Figure 3, Appendix A). The latter is characterized by a slightly lower amount of less sulfated HS GAG, a much higher RANKL/OPG ratio and the presence of inflammatory infiltrate (CD45).

### 3.2. Parameters of Regression Equation System

Pairing of variables and creation of the regression equation system was done using the STRING reactome network of factors under study as a template (Figure 4). The system consisted of 32 regression equations, of which 22 were SplR pairs, and 11 were MLRs. On average, the goodness of fit for SplRs was R^2^ = 0.99992 and for MLRs R^2^ = 0.86061 with *p* ≈ 0. A total of 12,342 knots were introduced for SplRs. A detailed list of statistical parameters for the first-level models can be found in Table 2.

### 3.3. The Effects of HPSE1 and Sdcs on the Expression/ECM Content of HS GAG—Module 1

The variables included in Module 1 are as follows: HPSE1 (main predictor), Sdc1–4 (intermediate predictors), and HS10E4 (main outcome; main predictor/link to module 2) (Figure 5A,B). The SplR pairs were modeled according to the well-documented role of HPSE1 as an endoglycosidase enzyme that cleaves HS GAG chains and facilitates shedding of the extracellular domains of Sdc [27,28,29]. Therefore, VKO or the absence of HPSE1 (−100% of baseline expression measured in diseased gingiva) is predicted to increase the expression of HS GAG and Sdcs. Conversely, overexpression of HPSE1 is predicted to lead to a gradual decrease in the expression of HS GAG and Sdcs. Peak levels at which overexpression of HPSE1 irreversibly downregulates the expression of HS GAG and Sdcs are predicted at 195% (2.95-fold) and 260% (3.6-fold) HPSE1 overexpression for the entire Sdcs complement and HS GAG, respectively. A nonlinear correlation between Sdcs expression and HS GAG is modeled (3rd order polynomial function). Sdcs are major cell surface receptors for HS GAG, and their presence partially overlaps with the expression of HS GAG in the ECM adjacent to cells. Thus, VKO of Sdcs is predicted to decrease the expression of HS GAG by nearly 50% (−2-fold), whereas the steady increase in the total amount of Sdcs positively correlates with the expression of HS GAG, up to a peak extrapolated at approximately 600% (7-fold) overexpression of Sdcs relative to baseline levels measured in diseased gingiva (Appendix A).

### 3.4. Biosynthesis of HS GAG and Feedback Loop between HS GAG and GAGosome Enzymes—Module 2

In Module 2, the balance between the ECM-located (external) HS GAG and the newly synthesized intracellular HS GAG was modeled. The latter is represented by four key enzymes responsible for the biosynthesis of HS GAG (EXT1-2, NDST1-2), forming the so-called GAGosome complex in the endoplasmic reticulum (ER) [30,31]. Therefore, the HS10E4 model from Module 1 was set as the main predictor with EXTs and NSDTs as intermediate predictors. Given the known functions of EXTs and NDSTs, HS10E4 and HS3G10 were set as the main outcomes for Module 2, representing total synthesized of HS GAG (HS10E4 + HS3G10) (Figure 5A,C). The assumption that quadruple VKO of EXTs and NDSTs leads to attenuation of HS GAG synthesis was also introduced. While EXTs have been shown to be essential for HS GAG biosynthesis, there is evidence that EXTs and NDSTs have overlapping functions [32,33,34]. Together with the relative concentrations of EXTs and NDSTs within the GAGosome complex, this has a major impact on the amount of newly synthesized HS GAG [35]. According to the models from Module 2, the amount of newly synthesized HS GAG varies nonlinearly (6th order polynomial function) as a function of differential expression of EXTs and NDSTs. The peak is predicted at 230% (3.3-fold) overexpression of HS GAG compared to baseline levels in diseased gingiva, after which GAGosome complex shutdown should occur (Figure 5C, Appendix A).

### 3.5. The Effects of SULF1-2 and SLC26A2 on Changes in Sulfation of HS GAG—Module 3

Module 3 was added as an open-ended side module to predict variation in sulfation of HS GAG in addition to its initial sulfation by the GAGosome enzymes modeled in Module 2. SLC26A2 and SULF1-2 were set as intermediate predictors with links to multiple variables (main predictors) from Module 1 and Module 2 based on the reconstruction of the STRING reactome network (Figure 4 and Figure 6A,B). SLC26A2 and SULF1-2 affect the sulfation of HS GAG in different ways, with SLC26A2 being a major supplier of sulfate ions (substrate), while SULF1-2 removes sulfate groups at specific sites in HS GAG chains (desulfation) [36,37,38,39]. As the main outcome, an artificial HS10E4 model variable was created and used to calculate the corrected values of HS GAG sulfation (expressed as a percentage) according to the simple formula (HS104/Total HS GAG) * 100. Depending on the differential expression of SLC26A2 and SULF1-2, the sulfation of HS GAG is predicted to vary below and above the value measured in healthy gingiva, which is on average 40% (1.4-fold) above the baseline value in diseased gingiva (Figure 6B, Appendix A).

### 3.6. Modulation of RANKL/OPG Ratio and CD45 Expression by HS GAG—Module 4

Modulation of the RANKL/OPG ratio and CD45 expression in the gingiva by HS GAG and related factors was modeled in Module 4. Thus, the RANKL/OPG ratio and CD45 were set as the main outcomes, with CD45 being the link variable to Module 1 (via the model variable HPSE1) (Figure 7A). In addition to STRING reactome network reconstruction (Figure 4), these assumptions were also considered for the arrangement of the main predictors (HS10E4, Sdc1, CD44) and the intermediate predictors (artificial variables OPGa–b; CD45a–c): (i) Sdc1 binds to OPG via HS GAG chains, making OPG less available to act as a decoy receptor for RANKL [40,41,42]. Thus, Sdc1 and HS GAG may shift the RANKL/OPG ratio in favor of RANKL, promoting bone resorption and inflammatory conditions; (ii) depletion of HS GAG content and lack of expression of Sdc1 and CD44 promote inflammatory conditions [43,44,45]; (iii) overexpression of Sdc1 and CD44 is also observed in some inflammatory conditions [18,46,47]. As predicted by the models in Module 4, VKO and overexpression of HS GAG could have opposite effects on the RANKL/OPG ratio and expression of CD45 (Figure 7B,C). However, the restoration of the RANKL/OPG ratio in healthy gingiva (approximately −80%, −5-fold compared with baseline in diseased gingiva) and CD45 expression (approximately −57%, −2.3-fold compared with baseline in diseased gingiva) could not be predicted for the entire range of HS GAG expression (VKO to 300%, 4-fold overexpression compared with baseline in diseased gingiva) (Appendix A).

### 3.7. Predicted Effects of Native HS GAG on Gingival Inflammatory Infiltrate and Alveolar Bone Formation/Degradation from the Overall Model

The expression profiles for HS GAG, the RANKL/OPG ratio, and CD45 were reproduced considering the cumulative network effect from Modules 1–4 (Figure 8). On this basis, it is predicted that decreasing presence of native HS GAG (from baseline level in diseased gingiva to VKO) decreases the RANKL/OPG ratio, while increasing expression of CD45. Similar effects on RANKL/OPG ratio and CD45 were predicted for overexpression of native HS GAG (200%, 3-fold overexpression from baseline in diseased gingiva). However, a steady increase in native HS GAG to these peaks is predicted to stabilize both the RANKL/OPG ratio and CD45 expression near baseline levels in diseased gingiva (Appendix A).

## 4. Discussion

The aim of this study was to evaluate the effects of native HS GAG on inflammatory infiltrate formation and alveolar bone resorption in severe periodontitis in humans using statistical modeling of data from IF staining of samples of human gingiva. The amount and structure (sulfation) of HS GAG differ between healthy and diseased gingiva, consistent with different expression profiles of proteins associated with biosynthesis and posttranslational modifications of HS GAG reported in our previous study [18]. The predictions of the statistical model imply that an increasing amount of native HS GAG in diseased gingiva could stabilize the spread of the inflammatory infiltrate and osteoclastogenic activity. On the other hand, the decreased presence (or absence) of native HS GAG could promote the spread of the inflammatory infiltrate while suppressing osteoclastogenic activity by lowering the RANKL/OPG ratio. However, we were unable to predict that increasing or decreasing the presence of native HS GAG in diseased gingiva could reduce formation of the inflammatory infiltrate and osteoclastogenic activity at levels observed in healthy gingiva.

The predictions of our model do not fully support the results of studies on the efficacy of supplementing degraded native HS GAG with its synthetic derivative (HS GAG mimetic) performed on experimental rodent models of periodontitis [16,48,49,50]. As reported, the administration of synthetic HS GAG was well tolerated by the animals, and the inflammation of soft periodontal tissues was significantly reduced. In addition, there were detectable signs of periodontal tissue regeneration after treatment with synthetic HS GAG. A proposed explanation was that synthetic HS GAG may be more resistant to fragmentation by degradative enzymes than native HS GAG due to its altered biochemical properties. However, the beneficial effects on periodontal tissue depended on the optimal dosage of synthetic HS GAG - it was found that administration of lower or higher doses may equally exacerbate the course of experimental periodontitis.

In animal models of periodontitis, researchers can replicate the clinical signs (gingivitis, alveolar bone resorption) but not the baseline conditions and the exact pathogenetic mechanism of periodontitis as it occurs in humans [51,52,53,54,55]. Therefore, the significant advantage of the model of periodontitis presented here is that it is based on the raw data obtained from human participants suffering from the real form of severe periodontitis. By combining the data from protein ontology databases and various regression techniques, a dynamic statistical model can be created, in which different roles of the factors under study can be incorporated, while the effects of their changing expression on the outcome(s) of interest can be replicated simultaneously. The arrangement of individual modules, quantification of IF in terms of spatial gradients, and selection of dominant functions for fitting segments in SplR pairs in first-level models were made under the assumption that key determinants of the biological roles of glycans, proteins, and other classes of molecules—hierarchical positioning within regulatory networks and spatial distribution within tissue—are just as important as the biochemical properties of individual molecules [56,57,58]. The problem with HS GAG and the proteins associated with HS GAG is that their roles in regulating pathogenetic mechanisms are as diverse as they are tissue- and species-specific. It is difficult to study them in humans, where experiments are not possible for ethical reasons. Therefore, the main advantage of this approach is that it shows how the effects of inhibition/absence or overexpression of specific factors (usually achieved by blocking antibodies or knockout/knockin of the corresponding genes in experimental animals) can be simulated in humans, at least virtually. It should be noted that these effects cannot be predicted simply by comparing the expression profiles of healthy and diseased tissue, nor can they be taken at face value from the studies on experimental animals [51,59].

There are several limitations to this study. Due to ethical considerations in research involving human participants, we could only create in silico model but could not experimentally test it in vivo. We were able to include only a limited number of variables while omitting many factors from the vast HS GAG reactome—these refer to enzymes responsible for HS GAG chain elongation (EXTL1-3), sulfation (NDST3, NDST4, 2-O- and 3-O-sulfotransferases), other cell-surface/ECM proteoglycans that bind HS GAG (agrin, HSPG2, glypicans), and many inflammatory mediators [60,61]. Therefore, certain aspects of the biosynthesis and posttranslational modifications of HS GAG, as well as its role in the formation and maintenance of gradients of inflammatory mediators, could not be modeled. As described previously, this limitation is due to the methodology for quantification and co-localization of IF from multiple markers. It relies on morphological similarity of serial histological sections, which limits the number of different markers whose IF signals can be co-localized (between 20 and 30 markers) [22]. There are additional tradeoffs from the application of IF/IHC and currently available commercial antibodies. For example, fragmentation of HS GAG and shedding of Sdcs cannot be visualized by IF/IHC staining with primary antibodies against HS GAG and Sdcs nor can they be extrapolated by overlapping their expression patterns with those of degradative enzymes such as HPSE1. Specifically, the antibodies against HPSE1 do not distinguish between active and inactive forms (proenzyme) of HPSE1, which means that the observed expression patterns of HPSE1 do not necessarily correspond to the actual enzymatic activity of HPSE1 [18,24]. It must be noted that the statistical model presented here is not deterministic and should be extended. This is a difficult task, both because of the inherent methodological limitations described above and because of the lack of data from experimental in vivo and in vitro studies from which the model assumptions are derived. Nevertheless, statistical modeling of human tissue data may provide an additional tool for evaluating potential molecular targets on which to base in vivo experiments for alternative treatment of chronic diseases such as severe periodontitis in both experimental animals and human participants. As with other chronic diseases, this may be of particular importance because their complex pathogenic mechanisms cannot be faithfully replicated in controlled laboratory settings.

## 5. Conclusions

Based on the predictions of our statistical model, we hypothesize that supplementation of native HS GAG may be of some benefit in the treatment of severe periodontitis in humans, but not as a sole treatment, as has been demonstrated in experimental animal models of periodontitis. The model supports the notion that intrinsic changes in the molecular structure of periodontal tissue are critical to the irreversible progression of severe periodontitis. This implies that effectively influencing the host (immune) response to treat severe periodontitis should go beyond the standard approach, which focuses on containing infectious agents from the microbial biofilm. Supplementing and/or targeting degraded ECM components with their synthetic derivatives (such as HS GAG mimetics) could be useful to improve the outcomes of standard treatment of severe forms of periodontitis, which usually falls short of periodontal tissue regeneration. However, more knowledge is needed about the complex biology of HS GAG and the factors associated with HS GAG.

## Figures and Tables

**Figure 1 bioengineering-09-00566-f001:**
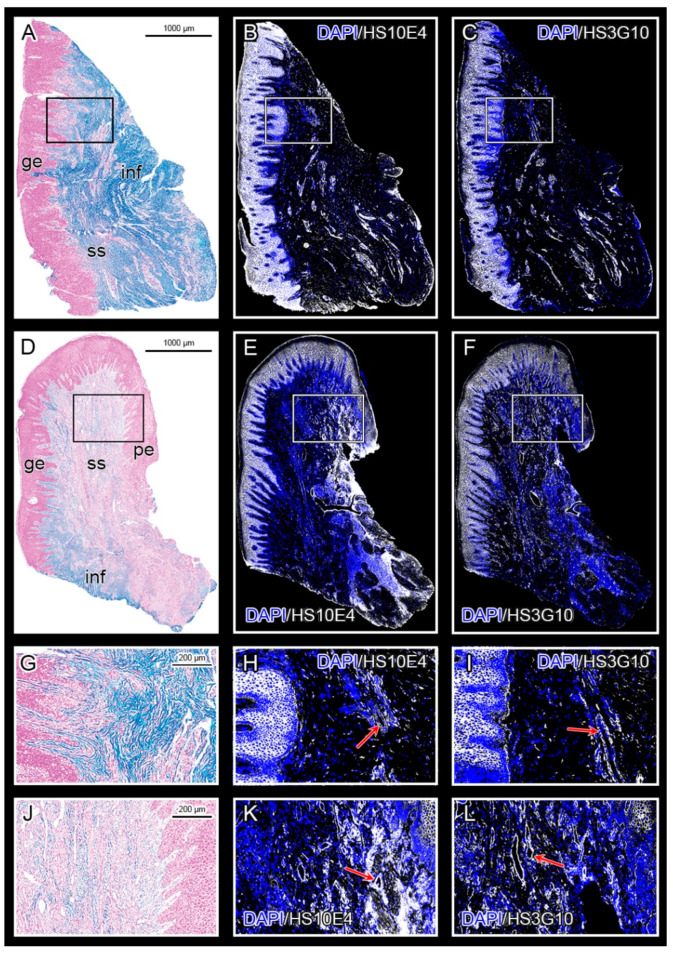
Panoramic images of healthy gingiva (control sample DK -NV2-ZK) (**A**–**C**) and diseased gingiva (severe periodontitis sample DK -SL17-CHP) (**D**–**F**); Alcian Blue staining (**A**,**D**); IF staining with primary antibodies against HS GAG, anti-HS10E4 (**B**,**E**) and anti-HS3G10 (**C**,**F**); The subepithelial stroma of healthy gingiva is rich in highly sulfated mucosubstances including GAGs (**A**, blue color), whereas their abundance is reduced in diseased gingiva (**D**). Reactivity with anti-HS10E4 (marks the sulfated regions of HS GAG chains) and anti-HS3G10 (marks the non-sulfated stumps of HS GAG) is visible mainly in the epithelial compartment of both healthy and diseased gingiva (white color; blue color—DAPI background staining); Reactivity to both antibodies is also seen in the subepithelial stroma (walls of blood vessels, inflammatory infiltrate), although it is more pronounced in the diseased gingiva. Magnified framed areas from Alcian Blue (**G**,**J**) and IF (**H**,**I**,**K**,**L**) panoramic images show the interface between the epithelial compartment and the subepithelial stroma with blood vessels (red arrows) strongly expressing HS GAG. (Magnification: ×10; Designations: gingival epithelium (**ge**), subepithelial stroma (**ss**), periodontal pocket epithelium (**pe**), and inflammatory infiltrate (**inf**). Image created in Adobe Photoshop 2020, version 21.2.0.

**Figure 2 bioengineering-09-00566-f002:**
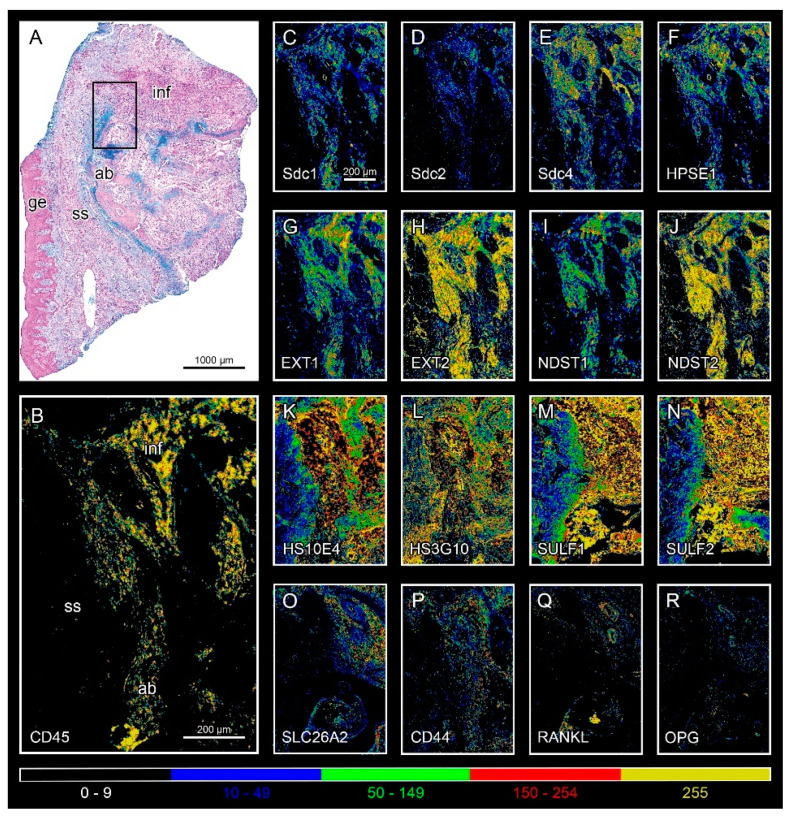
Expression of the studied markers in diseased gingiva (severe periodontitis sample DK-IP13-CHP) at the interface between noninfiltrated subepithelial stroma, alveolar bone, and inflammatory infiltrate (framed area on Alcian Blue panoramic image (**A**)). All markers are expressed in the area with different intensity (HS GAG cell surface receptors (**C**–**E,P)**; HS GAG (**K**,**L)**; HS GAG biosynthesis and degradation factors (**F**–**J**,**M**–**O)**; inflammatory cell marker (CD45) (**B)**; bone resorption/formation markers (RANKL, OPG) (**Q**,**R**); heatmap colors represent the intensity of IF signals: background (0–9 px GV; black), weak (10–49 px GV; blue), moderate (50–149 px GV; green), strong (150–254 px GV; red), and very strong intensity (255 px GV; yellow). (Magnification: ×10; Designations: gingival epithelium (**ge**), subepithelial stroma (**ss**), alveolar bone (**ab**), and inflammatory infiltrate (**inf**). The image was created with Adobe Photoshop 2020, version 21.2.0.

**Figure 3 bioengineering-09-00566-f003:**
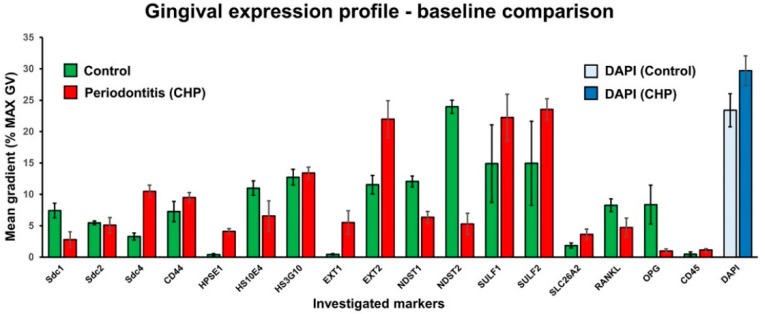
Bar-chart comparison of the expression profiles of the investigated markers in healthy (control) and diseased (severe periodontitis) gingiva samples. The expression of each marker is shown as the mean spatial gradient with standard deviation. The mean spatial gradients of DAPI staining are shown as reference values. The image was processed in Adobe Photoshop 2020, version 21.2.0.

**Figure 4 bioengineering-09-00566-f004:**
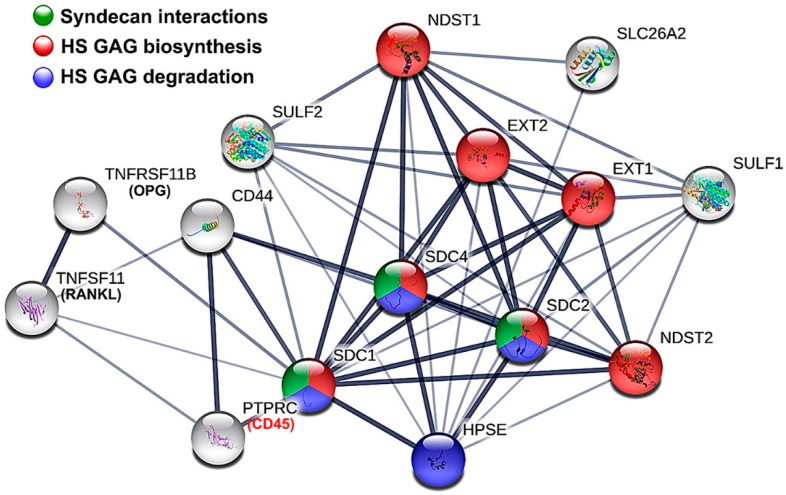
Functional protein network and enrichment from the STRING Reactome database. Network nodes represent the first shell and only 5 proteins from the second shell of interactors. Edges indicate confidence in functional and physical protein association. Colored nodes represent significantly enriched reactome pathways HSA-2022928 (HS GAG biosynthesis), HSA-2024096 (HS GAG degradation) and HSA-3000170 (Sdc interactions). Image processed in Adobe Photoshop 2020, version 21.2.0.

**Figure 5 bioengineering-09-00566-f005:**
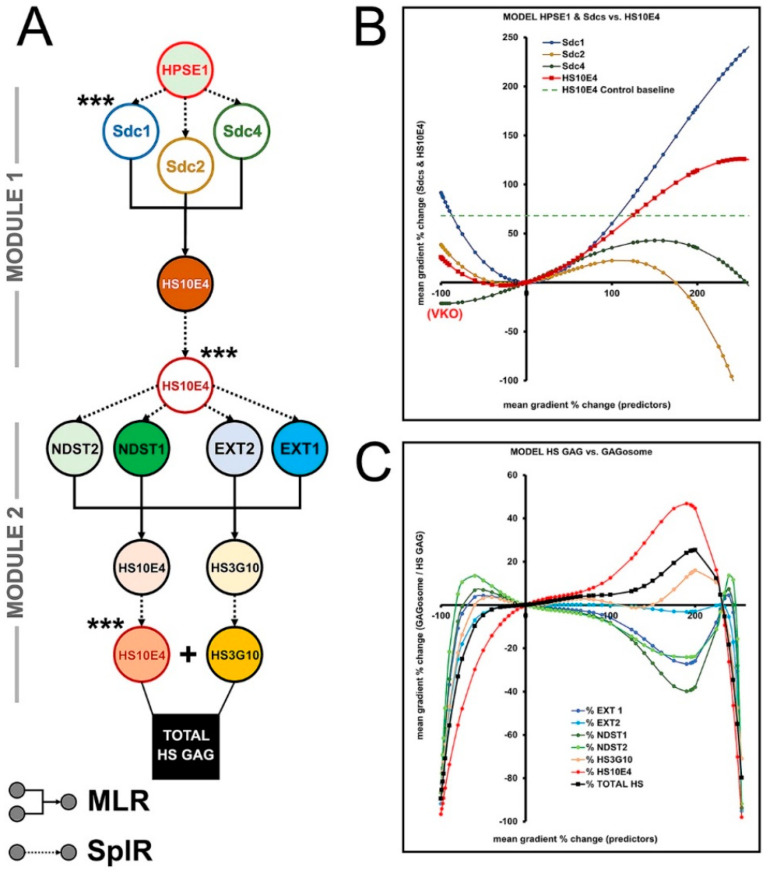
Modeling flowchart (**A**) and predictions (**B**,**C**) for the effects of HPSE1, Sdcs, and GAGosome enzymes on gingival HS GAG—Modules 1 and 2. Nodes in the flowchart represent individual variables. Note that the variables HS10E4 and HS3G10 were modeled iteratively. The total amount of HS GAG was calculated as the sum of HS10E4 and HS3G10 after the last iteration (Module 2). The values on the *x*–axis correspond to the difference between the input and baseline levels of the mean spatial gradients of the predictors measured in the diseased gingiva (severe periodontitis group). The values on the *y*–axis correspond to the difference between the predicted and baseline levels of the mean spatial gradients of the outcome variables measured in the diseased gingiva (severe periodontitis group). (Designations: multiple linear regression (**MLR**; furcated elbow arrows); spline regression (**SplR**; dotted arrows); virtual knockout (**VKO**); *** linking variable to succeeding module(s)). Image processed in Adobe Photoshop 2020, version 21.2.0.

**Figure 6 bioengineering-09-00566-f006:**
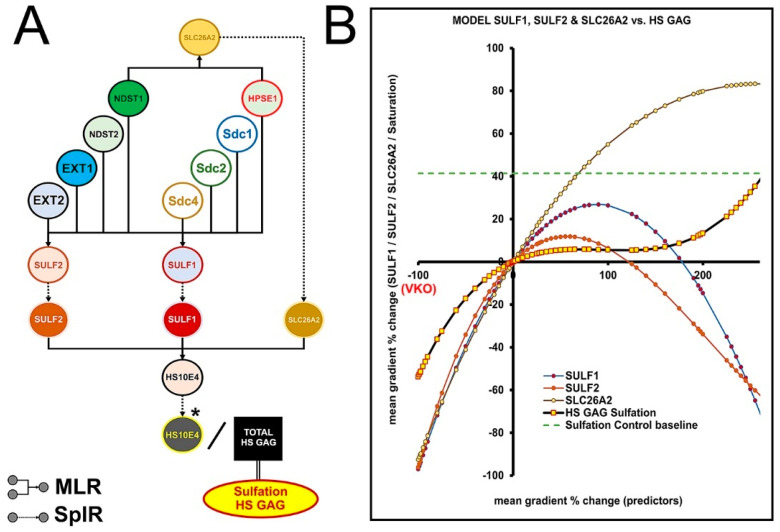
Modeling flowchart (**A**) and predictions (**B**) for the effects of SULF1–2 and SLC26A2 on sulfation of HS GAG – Module 3. Note that the artificial variable for HS10E4 was introduced to model the sulfation of HS GAG. The predicted values for the sulfation of HS GAG are below and above the baseline control level. (Designations: multiple linear regression (**MLR**; bifurcated elbow arrows); spline regression (**SplR**; dotted arrows); virtual knockout (**VKO**); * artificial variable). Image processed in Adobe Photoshop 2020, version 21.2.0.

**Figure 7 bioengineering-09-00566-f007:**
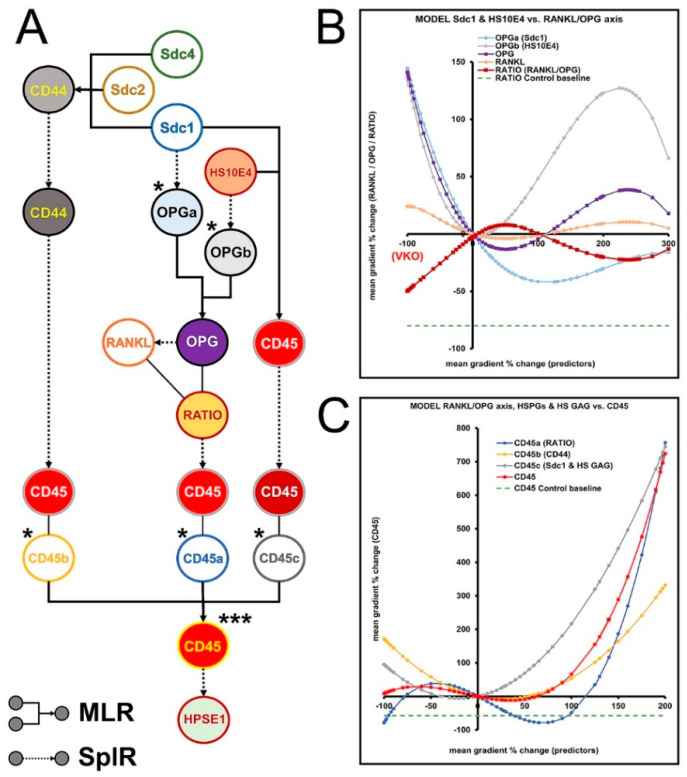
Modeling flowchart (**A**) and predictions for modulation of RANKL/OPG ratio (**B**) and CD45 expression (**C**) by HS GAG – Module 4. Artificial variables for OPG (OPGa–b) and CD45 (CD45a–c) are introduced to model the combined effect of a list of predictors (Sdcs, HS10E4, and CD44) on the main outcomes (RANKL/OPG ratio, CD45). (Designations: multiple linear regression (**MLR**; furcated elbow arrows); spline regression (**SplR**; dotted arrows); virtual knockout (**VKO**); * artificial variables; *** link variable to Module 1). Image processed in Adobe Photoshop 2020, version 21.2.0.

**Figure 8 bioengineering-09-00566-f008:**
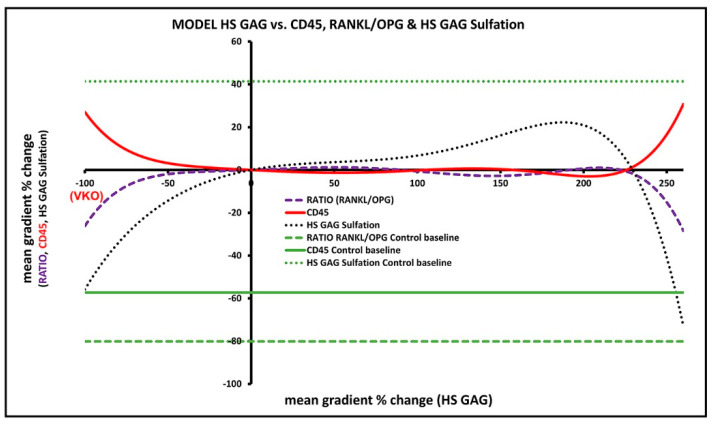
Overall model predictions for the effect of native HS GAG on inflammatory infiltrate in the gingiva and alveolar bone formation/degradation. Note the contrasting effects of altered expression of HS GAG (from **VKO** to 3.5–fold overexpression of HS GAG from baseline in diseased gingiva) on the RANKL/OPG ratio and CD45. However, for most of the overexpression range, HS GAG is predicted to stabilize the RANKL/OPG ratio and CD45 expression above control values in healthy gingiva. The image was processed in Adobe Photoshop 2020, version 21.2.0.

**Table 1 bioengineering-09-00566-t001:** Primary antibodies are used for IF staining of histological sections of gingiva.

Primary Antibody	Dilution	Description/Function
Mouse monoclonal anti-Sdc1 (ab34164) * ^a^	1:100	HSPGs; type 1 transmembrane proteins; cell surface receptors for HS GAG which participate in cell proliferation, cell migration and cell-matrix interactions; members of syndecan protein family
Rabbit polyclonal anti-Sdc2 (ab191062) * ^b^	1:200	
Rabbit polyclonal anti-Sdc4 (ab24511) * ^b^	1:100	
Rabbit polyclonal anti-EXT1 (ab126305) * ^b^	1:100	Exostosins; endoplasmic reticulum resident type 2 transmembrane glycosyltransferases; involved in biosynthesis of HS GAG (chain elongation step); members of the GAGosome complex
Rabbit polyclonal anti-EXT2 (ab102843) * ^b^	1:50	
Rabbit polyclonal anti-NDST1 (ab129248) * ^b^	1:50	Bifunctional enzymes involved in biosynthesis of HS GAG; catalyze N-deacetylation and N-sulfation of glucosamine monosaccharides of HS GAG (initial sulfation step); members of the GAGosome complex
Rabbit polyclonal anti-NDST2 (ab151141) * ^b^	1:100	
Mouse monoclonal anti-HS3G10 (370260-1) ^†^ ^c^	1:100	HS GAG marker; reacts with HS GAG 3G10 epitope generated after the digestion of HS GAG with heparinase III enzyme; binds to non-sulfated regions of HS GAG; does not react with other classes of GAGs
Mouse monoclonal anti-HS10E4 (370255-1) ^†^ ^c^	1:100	HS GAG marker; reacts with HS GAG 10E4 epitope which includes N-sulfated glucosamines; reactivity abolished after treatment with heparinase III enzyme; does not react with other classes of GAGs
Rabbit polyclonal anti-HPSE1 (ab85543) * ^b^	1:200	Heparanase 1 enzyme (endoglycosidase); degrades HS GAG at cell surface and within ECM into shorter oligosaccharide fragments; facilitates shedding of syndecans’ extracellular domains
Rabbit polyclonal anti-SULF1 (ab32763) * ^b^	1:200	Heparan sulfate 6-O-endo-sulfatases; selectively remove 6-O-sulfate groups from HS chains of HSPGs (HS GAG desulfation); modulate activity of HS GAG by altering binding sites for signaling molecules
Mouse monoclonal [2B4] anti-SULF2 (ab113405) * ^c^	1:25	
Rabbit polyclonal anti-SLC26A2 (ab238591) * ^b^	1:200	Transmembrane glycoprotein from the solute carrier protein family; involved in the transport of cellular sulfate ions essential for sulfation of HS GAG.
Rabbit polyclonal anti-CD44 (ab157107) * ^b^	1:500	Cell-surface receptor for hyaluronan; also binds HS GAG and interacts with Sdcs, collagens, matrix metalloproteinases; involved in cell- interactions, cell adhesion and migration
Mouse monoclonal anti-CD45 (PTPRC) (ab8216) * ^c^	1:200	Protein tyrosine phosphatase; type 1 transmembrane protein present in differentiated hematopoietic cells of myeloid and lymphoid lineage (common leukocyte antigen)
Rabbit polyclonal anti-RANKL (TNFSF11) (LS-B1425-0.05) ^‡ b^	1:200	Receptor activator of nuclear kappa-Β ligand (TNF receptor superfamily); type II membrane protein involved in bone remodeling and immune reactions; promotes differentiation and activation of osteoclasts and specific populations of immune cells
Rabbit polyclonal anti-OPG (TNFRSF11B) (ab73400) * ^b^	1:500	Osteoclastogenesis inhibitory factor (TNF receptor superfamily); decoy receptor for RANKL; inhibits osteoclastogenesis and bone resorption

Manufacturer: * Abcam plc, Cambridge, UK; ^†^ Seikagaku corp, Tokyo, Japan; ^‡^ LSBio inc, Seattle, WA, USA Secondary antibodies: ^a^ Donkey polyclonal anti-mouse Alexa Fluor 594 (ab150108); ^b^ Goat polyclonal anti-rabbit Alexa Fluor 488 (ab150077); ^c^ Donkey polyclonal anti-mouse Alexa Fluor 488 (ab150105); Secondary antibodies used at dilution 1:400.

**Table 2 bioengineering-09-00566-t002:** Regression parameters for SplR pairs and MLRs in first-level modeling of spatial gradients of investigated markers.

Model Parameters								
Predictors	Outcomes			R^2^	Df ^†^	FIT ^‡^	Knots	MODULE ^§^
	Marker	Mean *	Std. Error					
HPSE1	Sdc1	2.80123	0.03626	0.99912	1	SplR	545	1
	Sdc2	5.07486	0.00304	0.99999	1		579	
	Sdc4	10.49152	0.00395	0.99998	1		570	
Sdcs	HS10E4	6.54086	0.32258	0.98241	3	MLR	N/A	
HS10E4		6.54091	0.00663	0.99999	1	SplR	358	
HS10E4	EXT1	5.50899	0.00479	0.99999	1		226	2
	EXT2	21.97313	0.01919	0.99996	1		188	
	NDST1	6.346521	0.00357	0.99998	1		357	
	NDST2	5.29767	0.00963	0.99997	1		217	
EXTs, NDSTs	HS10E4	6.54091	0.31644	0.99795	4	MLR	N/A	
	HS3G10	13.41192	0.42029	0.99902	4			
HS10E4	HS10E4	6.54103	0.02772	0.99987	1	SplR	185	
HS3G10	HS3G10	13.41494	0.01222	0.99981	1		451	
HPSE1, Sdcs, EXTs, NDSTs	SULF1	22.24313	0.55832	0.97743	7	MLR	N/A	3
	SULF2	23.54117	0.34153	0.95915	7			
HPSE1, NDST1	SLC26A2	3.62778	0.53376	0.58391	2			
SULF1	SULF1	22.24361	0.01709	0.99998	1	SplR	642	
SULF2	SULF2	23.54061	0.01529	0.99992	1		730	
SLC26A2	SLC26A2	3.62767	0.00372	0.99998	1		646	
SULFs, SLC26A2	HS10E4	6.54181	0.37448	0.97629	3	MLR	N/A	
HS10E4	HS10E4	6.54103	0.00194	0.99999	1	SplR	1104	
Sdc1	OPG	0.97662	0.00158	0.99998	1	SplR	454	4
HS10E4		0.97665	0.00209	0.99996	1		437	
OPGa, OPGb		0.97671	0.00125	0.99998	2	MLR	N/A	
OPG	RANKL	4.72004	0.00279	0.99999	1	SplR	495	
RANKL/OPG RATIO	CD45	1.11445	0.00105	0.99999	1		776	
Sdcs	CD44	9.51209	0.64461	0.32535	3	MLR	N/A	
CD44	CD44	9.51218	0.00318	0.99998	1	SplR	712	
CD44	CD45	1.11444	0.00188	0.99993	1		888	
Sdc1, HS10E4	CD45	1.11445	0.13179	0.66525	2	MLR	N/A	
CD45	CD45	1.11445	0.00076	0.99998	1	SplR	683	
CD45a, CD45b, CD45c	CD45	1.11445	0.00049	0.99999	3	MLR	N/A	
CD45	HPSE1	4.10009	0.00481	0.99989	1	SplR	841	

* Mean value of investigated markers’ spatial gradients in diseased gingiva (periodontitis group) calculated from 7000 data points large T-D plots (n = 7000); values measured in % from px MAX GV; ^†^ degrees of freedom; ^‡^ Spline regression (SplR) or multiple regression (MLR) *p*-value for all models *p* ≈ 0 with confidence interval CI = 10^−8^; ^§^ Modules for the second- and third-level modeling corresponding to the reconstruction of functional protein network from STRING reactome databa.se.

## Data Availability

Data is contained within the article and Appendix A.

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
