# Peer review of "Heparan Sulfate Glycosaminoglycan Is Predicted to Stabilize Inflammatory Infiltrate Formation and RANKL/OPG Ratio in Severe Periodontitis in Humans"

_bioengineering, 2022, doi:10.3390/bioengineering9100566_

Round 1

Reviewer 1 Report

Dear Authors, 

Suggestions and Comments are mentioned in the pdf. 

Thank you, 

Regards.

Author Response

Structured abstract? According to the ”Instructions for authors”, this journal does not support structured abstracts.

Keywords in alphabetical order? Keywords are now placed in alphabetical order (highlighted in yellow color).

Any stats on prevalence of periodontitis? We have added some data on the global prevalence of severe forms of periodontitis in Introduction (1st paragraph, highlighted in yellow color).

What were the indices used for the measurement of clinical parameters? As stated in Materials and Methods (section 2.1. Samples procurement) we measured probing depth, gingival recession, clinical attachment level and full mouth plaque score. Diagnosis of severe periodontitis stage III and IV was based on those parameters.  

Inflammation goes up but osteoclast formation decreases? The inflammation may lead to ECM destruction, but inhibition of osteoclast formation should inhibit bone destruction, right? Is this correct? It is. However, here we predicted how changes in HS GAG expression affect the formation of inflammatory infiltrate and bone degradation through RANKL/OPG ratio which regulates osteoclastogenesis. HS GAG, just as factors associated with HS GAG, may have dual roles – promoting one process and simultaneously inhibiting the other. Or even promoting and inhibiting the same process depending on the local concentration of HS GAG. These assumptions were introduced in the model.     

Why not test the results in vitro? Neither 2D nor 3D cell cultures can replicate the complexity of tissue structure. One could use gingiva explants, but their viability is short. Besides, disease occurs in vivo so one cannot create the model of disease in vitro. Thus, we’re back to square one.

Add a heading called Conclusion and include additional information on translation of this research in vivo, future perspectives and clinical implications. According to the reviewer’s recommendation, parts of Discussion have been re-written and added Conclusion section added to elaborate on the future perspectives and clinical implications of this study.

Reviewer 2 Report

It is interesting to read the manuscript "Heparan Sulfate Glycosaminoglycan is Predicted to Stabilize Inflammatory Infiltrate Formation and RANKL/OPG Ratio in Severe Periodontitis in Humans". I appreciate the authors' efforts in conducting the study, which will be more beneficial to scholars working in the field. However, if the following changes are made and they are incorporated into the manuscript, it could be considered for publication in Bioengineering.

 1.    Avoid I/We/Our throughout the manuscript. Instead use “The present/current study”… I also advise authors to modify the title as "Heparan Sulfate Regulates Inflammatory Infiltrate Formation and RANKL/OPG Ratio in Severe Periodontitis in Humans".

2.    Line 18-20, revise the sentence. Since it is difficult to understand the objective.

3.    Describe the conclusion in more detail and offer suggestions for the reader. so that readers will understand the overall purpose of this research after reading the abstract.

4.    References 1, 2, and 3 appear to be very outdated. I advise including some recent references (s). For example https://doi.org/10.2147%2FJMDH.S374480

5.    Rewrite sentences in line 40-44. It is difficult to understand and perplexing as well.

6.    Line 91 - Detailed inclusion and exclusion criteria have been described previously. Despite the fact that it was mentioned in the previous article, the author must add the details here as well.

7.    Specify the date when each software tool utilised in this investigation was accessed.

8.    The results section has excellent writing. The lack of explanation in the discussion of each segment was another thing I noticed. Instead of giving more background information about the literature, I would advise the authors to refocus their discussion on each part so that it is evident how the research's findings fit into the larger context of what is happening right now with severe periodontitis.

9.    I suggest that authors separate the conclusion section and offer a critical justification on their findings and observations. As a result, everyone will understand the value of this research. Potential points of view must be covered in the conclusion as well. The importance of this study piece should be emphasised by the author.

Author Response

It is interesting to read the manuscript "Heparan Sulfate Glycosaminoglycan is Predicted to Stabilize Inflammatory Infiltrate Formation and RANKL/OPG Ratio in Severe Periodontitis in Humans". I appreciate the authors' efforts in conducting the study, which will be more beneficial to scholars working in the field. However, if the following changes are made and they are incorporated into the manuscript, it could be considered for publication in Bioengineering.

  1. Avoid I/We/Our throughout the manuscript. Instead use “The present/current study” … I also advise authors to modify the title as "Heparan Sulfate Regulates Inflammatory Infiltrate Formation and RANKL/OPG Ratio in Severe Periodontitis in Humans". Why is it wrong to use active form when reporting the findings? In fact, many authors prefer the active form instead of the passive form. In some sections of the original research article, the former is more appropriate than the latter (e.g., Methods, Results). In this manuscript we used both forms interchangeably, so we do not see the need to re-write the entire manuscript in passive form. Also, we must respectfully decline the recommendation to modify the title – the title should reflect the main finding of the study. If we modify the title according to the reviewer’s suggestion, we do not reveal anything new. In fact, we’d be restating the findings from previous experimental studies which were used as base assumptions for this study.   

  2. Line 18-20, revise the sentence. Since it is difficult to understand the objective. According to reviewer’s recommendation, sentence has been shortened (highlighted in green color).

  3. Describe the conclusion in more detail and offer suggestions for the reader. so that readers will understand the overall purpose of this research after reading the abstract. According to reviewer’s recommendation, we have added the concluding remark at the end of abstract (highlighted in green color).

  4. References 1, 2, and 3 appear to be very outdated. I advise including some recent references (s). For example https://doi.org/10.2147%2FJMDH.S374480 References 1-3 are related to some general facts about severe forms of periodontitis and are from studies published in 2017. and 2018. We would kindly ask reviewer to clarify why these refences are outdated and less appropriate than the recommended study by Selvaraj S et al. (2022) about the epidemiological factors of periodontal disease among south Indian adults which was performed on 288 participants from Tamil Nadu state in India?

  5. Rewrite sentences in line 40-44. It is difficult to understand and perplexing as well. We kindly ask reviewer to provide some suggestions/examples as we are not sure how to proceed on this point.

  6. Line 91 - Detailed inclusion and exclusion criteria have been described previously. Even though it was mentioned in the previous article, the author must add the details here as well. According to the reviewer’s recommendation, a short description of the inclusion and exclusion criteria has been added in Material and Methods (section 2.1. Samples procurement, 1st paragraph, highlighted in green color).

  7. Specify the date when each software tool utilised in this investigation was accessed. According to the reviewer’s suggestion, dates for the creation of each figure were added in figure legends (highlighted in green color).

  8. The results section has excellent writing. The lack of explanation in the discussion of each segment was another thing I noticed. Instead of giving more background information about the literature, I would advise the authors to refocus their discussion on each part so that it is evident how the research's findings fit into the larger context of what is happening right now with severe periodontitis. We understand the reviewer’s concern as we considered this in the original manuscript draft. However, if we were to debate each module in Discussion, the section would become super lengthy. Plus, we’d be repeating the statements provided in Results to support the specific assumptions for each module. Therefore, in Discussion we decided to focus on the main outcomes and predictions from the overall model in the context of the most relevant studies (ref. 16, 48-50).

  9. I suggest that authors separate the conclusion section and offer a critical justification on their findings and observations. As a result, everyone will understand the value of this research. Potential points of view must be covered in the conclusion as well. The importance of this study piece should be emphasised by the author. According to the reviewer’s recommendation, we have re-written the last paragraph of the Discussion and added separate Conclusion section (highlighted in green color).

Reviewer 3 Report

Dear authors, here goes a few concerns:

The Title and Abstract look acceptable.

In the Abstract I suggest the correction of the double-parenthesis on “… osteopro-25 tegerin (OPG)) …”

I recommend the keywords to be placed by alphabetic order.

I suggest the author to re-write their aim sentence. The objectives should be stated in a short, clear and direct sentence. And the objective are not clear, the authors state the goal is to evaluate the role, but can the authors be more specific on the parameters that constitute that role. Additionally, stating that the results will be discussed is not needed because that is obviously mandatory.

How was the sample size determined?

How were the patients selected? Random? Consecutive cases?

Was the methodology reviewed by a local ethics committee?

The results are extensive but well exposed.

In the Discussion I suggest the authors to debate the study strength, limitations, internal and external validity and further research. The rest of the theoretical support is fine.  

Author Response

Dear authors, here goes a few concerns:

The Title and Abstract look acceptable.

In the Abstract I suggest the correction of the double-parenthesis on “… osteopro-25 tegerin (OPG)) …” Double parenthesis on …OPG)) is not a typo. Please check the entire text within parentheses “…(represented by the ratio between…)

I recommend the keywords to be placed by alphabetic order. Keywords are now placed in alphabetical order.

I suggest the author to re-write their aim sentence. The objectives should be stated in a short, clear, and direct sentence. And the objective is not clear, the authors state the goal is to evaluate the role, but can the authors be more specific on the parameters that constitute that role. Additionally, stating that the results will be discussed is not needed because that is obviously mandatory. According to reviewer’s recommendation, aim sentence has been re-phrased (Introduction, highlighted in blue color). We have also removed the last sentence in Introduction about results being discussed.

How was the sample size determined? For details, please check Duplancic & Kero 2021. (ref. 22). Neither power analysis nor Mead’s equation (or similar procedures) for determining the sample size were used because they were not necessary. We should emphasize that the number of gingiva samples per group (n = 20) must not be mistaken with the sample size for regression analyses applied in this study. The actual sample size is derived from the output of 2D plots for spatial gradients (quantified IF staining) and is equal to n = 7000 (Table 2 legend: Dataset 1 – Regression matrices). P-hacking? Not at all. In this case, the object of analysis is NOT individual participant, but rather the marker – introducing the marker expression as a single value in regression model would cause significant information loss.   

How were the patients selected? Random? Consecutive cases? Participants were randomly selected. The screening of patients has been described in detail in Duplancic et al. 2019 (ref. 18). The inclusion criteria were 18+ years of age, good general health (no systemic diseases, healthy periodontal tissue (controls), generalized periodontitis stage III and IV (severe periodontitis group). The exclusion criteria were presence of periodontal abscess and endo-periodontal lesions in the vicinity of the sampling area, systemic diseases (e.g., diabetes mellitus), long-term medication, medical history of systemic antibiotic therapy within the last 6 months, pregnancy, alcohol, or drug consumption. In periodontitis group, gingival samples were obtained from a single tooth designated for extraction due to extensive damage of periodontal tissue. Demographic parameters of control and periodontitis group were also shown in Duplancic et al. 2019 (ref. 18).  

Was the methodology reviewed by a local ethics committee? Yes, it was. According to the journal’s instructions for authors, we have clearly stated that in Institutional Review Board Statement section.

In the Discussion I suggest the authors to debate the study strength, limitations, internal and external validity, and further research. The rest of the theoretical support is fine. The strengths and limitations of the study were presented in the 3rd and 4th paragraph of the Discussion section, respectively. Accordingly, recommendations for further research were proposed in the 4th paragraph of the Discussion and in Conclusion. Therefore, we kindly ask reviewer to specify additional points regarding the strength and limitations of the study that also need to be addressed. Considering the internal validity, we have provided the list of general assumptions for modeling in Material and Methods (section 2.5. Statistical analysis and modeling, 3rd paragraph), as well as the specific assumptions for each module in Results section. Once these were met, predictions were extrapolated and cross-referenced with control baseline levels (check Figures 3, 5-8; Supplementary Datasets 1 and 3). We have also added a clarification considering the external validity of model in the last paragraph of the Discussion section (highlighted in blue color).         

Round 2

Reviewer 2 Report

I appreciate the authors' efforts in making the necessary revisions to the manuscript in response to the given comments. Before it is considered for publication in Bioengineering, a few minor adjustments are needed.

1: Avoid I/We/Our throughout the manuscript. Instead use “The present/current study” … I also advise authors to modify the title as "Heparan Sulfate Regulates Inflammatory Infiltrate Formation and RANKL/OPG Ratio in Severe Periodontitis in Humans". 

Author’s response: Why is it wrong to use active form when reporting the findings? In fact, many authors prefer the active form instead of the passive form. In some sections of the original research article, the former is more appropriate than the latter (e.g., Methods, Results). In this manuscript we used both forms interchangeably, so we do not see the need to re-write the entire manuscript in passive form. Also, we must respectfully decline the recommendation to modify the title – the title should reflect the main finding of the study. If we modify the title according to the reviewer’s suggestion, we do not reveal anything new. In fact, we’d be restating the findings from previous experimental studies which were used as base assumptions for this study.

Comment from Reviewer: Yes. I concurred with some of the authors' arguments. However, why I have been mentioned to remove I/We/Our is because readers could see such writing as being subjective, but science is all about objectivity. Further, in the title, I'm not clear what aspect of novelty in the recommended title has been eliminated. Instead of the title provided by the authors, the recommended title better captures the overall study. “Heparan Sulfate Glycosaminoglycan is Predicted to Stabilize” has been modified to "Heparan Sulfate Regulates”

 2. Line 18-20, revise the sentence. Since it is difficult to understand the objective.

Author’s response: According to reviewer’s recommendation, sentence has been shortened (highlighted in green color).

 Comment from Reviewer: It is now in an acceptable form.

3.     Describe the conclusion in more detail and offer suggestions for the reader. so that readers will understand the overall purpose of this research after reading the abstract. 

Author’s response: According to reviewer’s recommendation, we have added the concluding remark at the end of abstract (highlighted in green color).

Comment from Reviewer: It is now in an acceptable form.

 4.     References 1, 2, and 3 appear to be very outdated. I advise including some recent references (s). For example https://doi.org/10.2147%2FJMDH.S374480 

Author’s response: References 1-3 are related to some general facts about severe forms of periodontitis and are from studies published in 2017. and 2018. We would kindly ask reviewer to clarify why these refences are outdated and less appropriate than the recommended study by Selvaraj S et al. (2022) about the epidemiological factors of periodontal disease among south Indian adults which was performed on 288 participants from Tamil Nadu state in India?

Comment from Reviewer: Not necessary to include the reference that I mentioned in my previous comment. It is one of several recent articles about periodontitis that have been published recently. That's why I suggested.  Since the author is providing some general information concerning severe forms of periodontitis, it is unclear why they were unable to find any relevant studies published in the last five years (2018–2022) where they could discover the most recent statistics.

5.     Rewrite sentences in line 40-44. It is difficult to understand and perplexing as well. We kindly ask reviewer to provide some suggestions/examples as we are not sure how to proceed on this point.

Comment from Reviewer: Authors can verify the following and, if I mentioned it right, can be adopted.

“Studies using Sdcs knockout mice (KO) have demonstrated that the absence of individual Sdc in a tissue can enhance inflammation, disrupt wound healing, and enhance resistance to microbial infections. Further, when Sdcs are overexpressed, multiple roles for them as cellular process modulators have been found in pathological conditions.”

  1. Line 91 - Detailed inclusion and exclusion criteria have been described previously. Even though it was mentioned in the previous article, the author must add the details here as well. 

Author’s response: According to the reviewer’s recommendation, a short description of the inclusion and exclusion criteria has been added in Material and Methods (section 2.1. Samples procurement, 1st paragraph, highlighted in green color).

Comment from Reviewer: It is now in an acceptable form.

  1. Specify the date when each software tool utilised in this investigation was accessed. 

Author’s response: According to the reviewer’s suggestion, dates for the creation of each figure were added in figure legends (highlighted in green color).

Comment from Reviewer: It is now in an acceptable form.

  1. The results section has excellent writing. The lack of explanation in the discussion of each segment was another thing I noticed. Instead of giving more background information about the literature, I would advise the authors to refocus their discussion on each part so that it is evident how the research's findings fit into the larger context of what is happening right now with severe periodontitis. 

Author’s response: We understand the reviewer’s concern as we considered this in the original manuscript draft. However, if we were to debate each module in Discussion, the section would become super lengthy. Plus, we’d be repeating the statements provided in Results to support the specific assumptions for each module. Therefore, in Discussion we decided to focus on the main outcomes and predictions from the overall model in the context of the most relevant studies (ref. 16, 48-50).

Comment from Reviewer: It is now in an acceptable form.

  1. I suggest that authors separate the conclusion section and offer a critical justification on their findings and observations. As a result, everyone will understand the value of this research. Potential points of view must be covered in the conclusion as well. The importance of this study piece should be emphasised by the author.

Author’s response: According to the reviewer’s recommendation, we have re-written the last paragraph of the Discussion and added separate Conclusion section (highlighted in green color).

 Comment from Reviewer: It is now in an acceptable form.

Author Response

We would like to thank the reviewer 2 for clarification of specific points from the previous round of revision. Here are our responses:

I appreciate the authors' efforts in making the necessary revisions to the manuscript in response to the given comments. Before it is considered for publication in Bioengineering, a few minor adjustments are needed.

1: Avoid I/We/Our throughout the manuscript. Instead use “The present/current study” … I also advise authors to modify the title as "Heparan Sulfate Regulates Inflammatory Infiltrate Formation and RANKL/OPG Ratio in Severe Periodontitis in Humans". 

Author’s response: Why is it wrong to use active form when reporting the findings? In fact, many authors prefer the active form instead of the passive form. In some sections of the original research article, the former is more appropriate than the latter (e.g., Methods, Results). In this manuscript we used both forms interchangeably, so we do not see the need to re-write the entire manuscript in passive form. Also, we must respectfully decline the recommendation to modify the title – the title should reflect the main finding of the study. If we modify the title according to the reviewer’s suggestion, we do not reveal anything new. In fact, we’d be restating the findings from previous experimental studies which were used as base assumptions for this study.

Comment from Reviewer: Yes. I concurred with some of the authors' arguments. However, why I have been mentioned to remove I/We/Our is because readers could see such writing as being subjective, but science is all about objectivity. Further, in the title, I'm not clear what aspect of novelty in the recommended title has been eliminated. Instead of the title provided by the authors, the recommended title better captures the overall study. “Heparan Sulfate Glycosaminoglycan is Predicted to Stabilize” has been modified to "Heparan Sulfate Regulates”

As we stated before, active form and passive form are related to writing style and have nothing to do with being objective or subjective. If we performed staining of histological slides, why is it wrong to say: “We stained histological slides”? Additionally, being objective means to base conclusions from the correct interpretation of the data with respect to methodological limitations. Considering the recommended changes of title, every aspect of novelty is eliminated if we formulate the title “Heparan Sulfate Regulates…”. It is well known that HS regulates inflammation and bone resorption, that’s what motivated us to do this study. We also provided many references to support that. But how HS regulates inflammation and bone metabolism in human periodontal tissue (is it pro- or anti-inflammatory? is it osteoclastogenic?) is not known because the effects of HS are tissue- and species-specific. So, the fact that HS can stabilize formation of inflammatory infiltrate and RANKL/OPG ratio is the novelty.  

  1. Line 18-20, revise the sentence. Since it is difficult to understand the objective.

Author’s response: According to reviewer’s recommendation, sentence has been shortened (highlighted in green color).

 Comment from Reviewer: It is now in an acceptable form.

  1. Describe the conclusion in more detail and offer suggestions for the reader. so that readers will understand the overall purpose of this research after reading the abstract. 

Author’s response: According to reviewer’s recommendation, we have added the concluding remark at the end of abstract (highlighted in green color).

Comment from Reviewer: It is now in an acceptable form.

  1. References 1, 2, and 3 appear to be very outdated. I advise including some recent references (s). For example https://doi.org/10.2147%2FJMDH.S374480 

Author’s response: References 1-3 are related to some general facts about severe forms of periodontitis and are from studies published in 2017. and 2018. We would kindly ask reviewer to clarify why these refences are outdated and less appropriate than the recommended study by Selvaraj S et al. (2022) about the epidemiological factors of periodontal disease among south Indian adults which was performed on 288 participants from Tamil Nadu state in India?

Comment from Reviewer: Not necessary to include the reference that I mentioned in my previous comment. It is one of several recent articles about periodontitis that have been published recently. That's why I suggested.  Since the author is providing some general information concerning severe forms of periodontitis, it is unclear why they were unable to find any relevant studies published in the last five years (2018–2022) where they could discover the most recent statistics.

The opening paragraph in Introduction provides general facts about severe periodontitis which haven’t changed. The disease is still highly prevalent, and it is public health problem. So, it made sense to cite several studies which compiled results from numerous epidemiological studies. Besides, one of these references fits within 2018-2022 time frame.

  1. Rewrite sentences in line 40-44. It is difficult to understand and perplexing as well. We kindly ask reviewer to provide some suggestions/examples as we are not sure how to proceed on this point.

Comment from Reviewer: Authors can verify the following and, if I mentioned it right, can be adopted.

“Studies using Sdcs knockout mice (KO) have demonstrated that the absence of individual Sdc in a tissue can enhance inflammation, disrupt wound healing, and enhance resistance to microbial infections. Further, when Sdcs are overexpressed, multiple roles for them as cellular process modulators have been found in pathological conditions.”

According to reviewer’s recommendation, we have modified the sentences like this (highlighted in grey color):

“Studies on Sdcs knockout mice (KO) have shown that the absence of individual Sdcs in tissues can promote inflammation, disrupt wound healing, and increase resistance to microbial infections [7], [8]. In addition, overexpression of Sdcs has been reported in various pathological conditions. Therefore, Sdcs may be characterized as modulators of cellular processes.”

  1. Line 91 - Detailed inclusion and exclusion criteria have been described previously. Even though it was mentioned in the previous article, the author must add the details here as well. 

Author’s response: According to the reviewer’s recommendation, a short description of the inclusion and exclusion criteria has been added in Material and Methods (section 2.1. Samples procurement, 1st paragraph, highlighted in green color).

Comment from Reviewer: It is now in an acceptable form.

  1. Specify the date when each software tool utilised in this investigation was accessed. 

Author’s response: According to the reviewer’s suggestion, dates for the creation of each figure were added in figure legends (highlighted in green color).

Comment from Reviewer: It is now in an acceptable form.

  1. The results section has excellent writing. The lack of explanation in the discussion of each segment was another thing I noticed. Instead of giving more background information about the literature, I would advise the authors to refocus their discussion on each part so that it is evident how the research's findings fit into the larger context of what is happening right now with severe periodontitis. 

Author’s response: We understand the reviewer’s concern as we considered this in the original manuscript draft. However, if we were to debate each module in Discussion, the section would become super lengthy. Plus, we’d be repeating the statements provided in Results to support the specific assumptions for each module. Therefore, in Discussion we decided to focus on the main outcomes and predictions from the overall model in the context of the most relevant studies (ref. 16, 48-50).

Comment from Reviewer: It is now in an acceptable form.

  1. I suggest that authors separate the conclusion section and offer a critical justification on their findings and observations. As a result, everyone will understand the value of this research. Potential points of view must be covered in the conclusion as well. The importance of this study piece should be emphasised by the author.

Author’s response: According to the reviewer’s recommendation, we have re-written the last paragraph of the Discussion and added separate Conclusion section (highlighted in green color).

 Comment from Reviewer: It is now in an acceptable form.

Reviewer 3 Report

Dear authors, I have no further concerns.

Author Response

We would like to thank reviewer 3 for accepting revisions from round 1.